# Saturated Fatty Acid-Based In Situ Forming Matrices for Localized Antimicrobial Delivery

**DOI:** 10.3390/pharmaceutics12090808

**Published:** 2020-08-25

**Authors:** Takron Chantadee, Wichai Santimaleeworagun, Yaowaruk Phorom, Thawatchai Phaechamud

**Affiliations:** 1Department of Pharmaceutical Technology, Faculty of Pharmacy, Silpakorn University, Nakhon Pathom 73000, Thailand; 2Department of Pharmacy, Faculty of Pharmacy, Silpakorn University, Nakhon Pathom 73000, Thailand; santimaleeworag_w@su.ac.th; 3Secretary Office of Faculty, Faculty of Pharmacy, Silpakorn University, Nakhon Pathom 73000, Thailand; phorom_y@su.ac.th; 4Natural Bioactive and Material for Health Promotion and Drug Delivery System Group (NBM Group), Faculty of Pharmacy, Silpakorn University, Nakhon Pathom 73000, Thailand

**Keywords:** saturated fatty acid, antisolvent process, local drug delivery, vancomycin HCl, in situ forming matrix

## Abstract

In recent years, the world has faced the issue of antibiotic resistance. Methicillin-resistant *Staphylococcus aureus* (MRSA) is a significant problem in various treatments and control of infections. Biocompatible materials with saturated fatty acids of different chain lengths (C_8_–C_18_) were studied as matrix formers of localized injectable vancomycin HCl (VCM)-loaded antisolvent-induced in situ forming matrices. The series of fatty acid-based in situ forming matrices showed a low viscosity (5.47–13.97 cPs) and pH value in the range of 5.16–6.78, with high injectability through a 27-G needle (1.55–3.12 N). The preparations exhibited low tolerance to high concentrations of KH_2_PO_4_ solution (1.88–5.42% *v/v*) and depicted an electrical potential change during phase transformation. Their phase transition and matrix formation at the microscopic and macroscopic levels depended on the chain length of fatty acids and solvent characteristics. The VCM release pattern depended on the nucleation/crystallization and solvent exchange behaviors of the delivery system. The 35% *w/v* of C_12_–C_16_ fatty acid-based in situ forming matrix prolonged the VCM release over seven days in which C_12_, C_14_, C_16_ –based formulation reached 56, 84, and 85% cumulative drug release at 7th day. The release data fitted well with Higuchi’s model. The developed formulations presented efficient antimicrobial activities against standard *S. aureus*, MRSA, *Escherichia coli*, and *Candida albicans*. Hence, VCM-loaded antisolvent-induced fatty acid-based in situ forming matrix is a potential local delivery system for the treatment of local Gram-positive infection sites, such as joints, eyes, dermis of surgery sites, etc., in the future.

## 1. Introduction

The in situ forming system has emerged as a useful drug delivery system owing to its unique behavior of self-transformation at the target site. A liquid dosage form becomes semisolid or solid via various mechanisms, such as thermal trigger [1,2], pH change [3], solvent exchange [4], and chemical reaction [3,5]. The system has been focused on various application sites, such as the eyeball [6], periodontal pocket [7], articular joint [8], and muscle [9]. The solvent exchange or antisolvent processes are one of the simple mechanisms that suit any life form, given that the biological fluid is located in almost every part of the being. The processes involve the migration of solvent outward, whereas the nonsolvent moves inward to the system, resulting in the separation of the matrix former by the antisolvent process [10,11]. In general, nonsolvent refers to a biological fluid, that is, an aqueous fluid. Most hydrophobic or aqueous insoluble polymers, such as poly(lactide-co-glycolide), ethylcellulose, bleached shellac, and Eudragit RS, have been employed as skeleton formers [11,12]. Not only polymers but also lipids are of interest for antisolvent-induced in situ forming system development. Cholesterol has been used as a matrix former in in situ forming gel to deliver doxycycline hyclate to the periodontal pocket for periodontitis treatment [13]. Phospholipid-based phase transition gel was developed for the prolonged release of ropivacaine [14]. Fatty acids are synthesized/metabolized in the human body and found in our food [15,16,17,18]; they are known for their biocompatible and inert properties [18,19]. They have also been widely used in the pharmaceutical and cosmetical fields [20,21]. The different physicochemical properties of saturated fatty acids result in various outcomes that can be applied for different purposes. For example, the use of a high-molecular-weight (MW) saturated fatty acid promotes the release-sustaining capability [22] and minimizes drug/fatty acid interaction [23], reducing the thermal conductivity of formulas [24]. Six saturated fatty acids that carry 8–18 carbon atoms were selected to be studied as core components for fatty acid-based in situ forming matrix in this investigation. Caprylic acid (CPL), capric acid (CPR), lauric acid (LAU), myristic acid (MYR), palmitic acid (PAL), and stearic acid (STR) were used as the matrix formers.

The resistance to antibiotics of Gram-positive bacteria has become a global problem; particularly, methicillin-resistant *Staphylococcus aureus* (MRSA) is a major problem in the treatment and control of staphylococcal infections [25,26]. MRSA surgical site infections have reached a high cost and mortality [27]. Given the imminence of bacteria resistance, potent antibiotics may be an optimal solution. Vancomycin HCl (VCM), a bulky hydrophilic drug, has been widely used to treat infections caused by Gram-positive bacteria, especially *Staphylococcus aureus* (including MRSA). VCM has activity against *Streptococcus* spp., *Enterococcus* spp., *Staphylococcus* spp., and *Clostidium* spp. [28]. The half maximal inhibitory concentration (IC_50_)/minimum inhibition concentration(MIC) of vancomycin HCl against *S. aureus*, MRSA, and *E. faecalis* were 0.37/0.625, 0.67/0.125, and 0.73/1.25 µg/mL, respectively [29]. While the MIC values of vancomycin HCl against beta-hemolytic streptococci, which include *S. pyogenes*, *S. agalactiae*, *S. dysgalactiae*, *S. anginosus* group, Group C streptococci, and Group G streptococci, were in the range of ≤0.015 to 1 µg/mL [30]. VCM is the drug of choice for MRSA infection [31]. This drug can be applied through various administration routes, such as intravenous, ophthalmic, oral cavity, and intra-articular means, which rely on infection sites [32,33]. The intravenous route is usually selected owing to its low availability. In most cases, infections are located at particular sites, such as joints [34,35], eye [36], dental area [37,38], and any postsurgery site [39]. Local/targeted drug delivery systems can provide sufficient drug level at the site with low concentrations in the systemic system, resulting in less side effects [40]. The therapeutic period and level of VCM at the target site are crucial factors for achieving an effective treatment [33]. Meanwhile, in situ forming systems operate via an antisolvent mechanism that can prolong the release of VCM. The VCM-loaded in situ forming systems can be developed for the treatment of various infection sites. Recently, LAU and the mixture of LAU/STR were employed for the in situ forming matrix development, which was aimed at their use for periodontitis and postoperative knee infection [41,42]. These in situ forming matrices using LAU and STR showed different properties, such as matrix formation behavior, matrix formation rate, and other backstage properties. However, the following phenomena were not revealed: pattern of aqueous influx during in situ formation process; the initial matrix formation at the boundary of formulation/simulated biological stage; effect of various aqueous environments; and effect of MW of saturated fatty acid on in situ forming matrix.

Therefore, this study aimed to apply various saturated fatty acids (C_8_–C_18_) as matrix formers in an in situ forming matrix to prolong the release of VCM and investigated the effect of their chain length on in situ formation via the antisolvent process. Solutions containing various fatty acids in dimethyl sulfoxide (DMSO) and *N*-methyl-2-pyrrolidone (NMP) were prepared. The following physicochemical properties and their behaviors, which were included in the new experiments for the abovementioned phenomena, were evaluated: pH, density, viscosity, in vitro matrix formation behavior, matrix formation rate, interfacial behavior, electrical potential change during phase separation, and injectability. Moreover, the influence of KH_2_PO_4_ on the formulations, in vitro drug release, aqueous phase migration behavior, topography of transformed matrix, and antibacterial activities were investigated.

## 2. Materials and Methods

### 2.1. Materials

CPL (≥99%, Batch No. 289D1190515A), CPR (≥99%, Batch No. 219D161222A), LAU (≥99%, Batch No. 229F170S08A), MYR (≥99%, Batch No. FPLK437 × 4S), and PAL (≥99%, Batch No. 668F180902D), which were procured from Pacific Oleochemicals, Pasir Gudang, Malaysia, and STR (≥98, Batch No. 3452018), which was purchased from CT Chemical, Bangkok, Thailand, were used as matrix formers. VCM (≥80%, Lot No. WXBB5169V, Sigma-Aldrich, Co., St. Louis, MO, USA) was used as a model drug. NMP (≥99.5%, Lot No. A0251390, Fluka, Ballwin, MO, USA) and DMSO (≥99.9%, Lot No. 453035, Fluka, Buchs, Switzerland) were used as solvents. Muller Hinton agar (Lot No. 1005843, Oxoid, Hampshire, UK) was used as medium for the antimicrobial tests. *S. aureus* ATCC 43300, *S. aureus* DMST 6532, *S. aureus* ATCC 25923, *E. coli* ATCC 8739, and *C. albicans* ATCC 17100 were used as test microbials. Potassium dihydrogen orthophosphate (≥99%, Lot No. E23W60, Ajax Finechem, New South Wales, Australia) and sodium hydroxide (≥97%, Lot No. AF 310204, Ajax Finechem, New South Wales, Australia) were used as ingredients of phosphate buffer (PB) with pH 7.4 to simulate the physiological fluid. Agarose (Lot No. H7014714, Vivantis, Selangor Darul Ehsan, Malaysia) and sodium–fluorescein (Lot No. MXCG7851, Sigma-Aldrich, St. Louis, MO, USA) were used for the analysis of matrix formation behavior and solvent diffusion.

### 2.2. Preparation of In Situ Forming Matrix

The saturated fatty acids were dissolved in NMP and DMSO, and the volume was adjusted to obtain 35% *w/v* fatty acid or 1.75 M. The mixtures were stirred for 24 h until clear solutions were obtained. The 1% *w/v* VCM in selected formulations was prepared by dissolving VCM in 35% *w/v* of fatty acid using DMSO as a solvent. Table 1 shows the components of each formulation.

### 2.3. Evaluation of In Situ Forming Matrix Systems

#### 2.3.1. pH, Density and Viscosity Evaluation

The pH values of all formulas were determined using a pH meter (Seven Compact, Mettler Toledo, Vernon Hills, IL, USA) (*n* = 3). The density was measured using a pycnometer (Densito 30PX, Mettler Toled Ltd., PortableLab TM, East Bunker Ct Vernon Hills, IL, USA) (*n* = 3). Viscosity and shear stress were conducted by using a viscometer (Brookfield Engineering Laboratories Inc., Middleborough, MA, USA) with a CP-40 spindle (*n* = 3) at 25 °C at a shear rate of 250 s^−1^.

#### 2.3.2. In Vitro Matrix Formation Behavior

The formation behavior of the prepared systems with macro-level observation was tested by injection of 0.5 mL preparation through an 18-gauge needle into a 5 mL PB (pH 7.4) and photographed them onto a black background using a digital camera (Samsung ST50, Suwon, Korea) at various time points (0, 1, and 5 min).

#### 2.3.3. Matrix Formation Rate Investigation (Cross-Sectional View)

Cross-sectional view matrix formation experiment was set up to determine the matrix formation rate. The 0.6% *w/w* agarose was prepared. Weighted agarose was dissolved in boiling PB and 10 mL of the obtained solution was poured into Petri dishes (diameter: 4.5 cm). At the centre of the settled gels, a cylindrical well (diameter: 6 mm) was made and subsequently filled with 150 µL formulation. The matrix formation was captured using a stereoscope (Motic SMZ-171 Series, Motic Asia, Kowloon, Hong Kong) at different time intervals (1, 3, 5, 10, and 15 min). The distance between the rim of agarose and the inner area of the matrix was measured using Motic SMZ-171 Series software (Motic Asia, Kowloon, Hong Kong) for five zones, with each measurement performed in triplicate. The rate of matrix formation was also calculated (*n* = 5).

#### 2.3.4. Interfacial Phenomenon during the Initial Experimental Period

A 0.6% *w/v* agarose solution was prepared as mentioned above. The obtained solution was poured onto a glass slide with a 2 mm-thick agarose. Then, 100 µL formulation was dropped at the boundary of the settled agarose. The interfacial event was time-lapse captured under an inverted microscope with 100× magnification (TE-2000U, Nikon, Kaw, Japan).

#### 2.3.5. Electrical Potential Difference During Phase Transformation

The electrical potential change was investigated using a conductometer (SevenCompact, Mettler Toledo, Ct Vernon Hills, IL, USA). Then, 4 mL samples were placed in the test tube. Next, 50 μL deionized water was added at each interval and the liquid sample was stirred with a vortex. Deionized water was added until a turbid liquid was observed (separation point). The voltage of deionized water was 371.2 ± 1.3 mV with conductivity of 0.1 µs/cm.

#### 2.3.6. Influence of KH_2_PO_4_ on Water Tolerance

The capacity of formulations to endure phase transformation was investigated by using the titration method. This experiment was set up to magnify the environmental factor influencing the water tolerance property. The high amount of titrated water (% *v/v*) at the separation point represents high water tolerance [42]. A total of 10 mL of each formula was filled into a 50 mL Erlenmeyer flask as the titrand and then titrated with deionized water and various concentrations of KH_2_PO_4_ solutions (12.5, 25, 50, 100, and 200 mM) until the clear solution turned turbid (separation point). The dropping rate of titration was 0.05 mL/s. During titration, the formulation was swirled rigorously at 150 rpm using a magnetic stirrer. The amount of titrant required for transformation as a cloud point was recorded and the water amount (% *v/v*) was calculated using Equation (1).
(1)%Water amount = Amount of titrant (mL)10+Amount of titrant (mL)×100

#### 2.3.7. Injectability Evaluation

The difficulty of injection of formulations was examined using a texture analyzer in the compression mode (TA.XT plus, Stable Micro Systems, Godalming, UK), through a 1 mL syringe/27-G needle when the upper probe of the instrument moved downwards at a constant speed (1.0 m s^−1^) with a constant force of 0.1 N (*n* = 3). A low force or work of injection indicated good injectability. The Brookfield DV-III Ultra programmable rheometer (Brookfield Engineering Laboratories Inc, Middleboro, MA, USA) was employed for measurement of rheological character of formula at 25 °C (*n* = 3). The flow parameters were calculated using an exponential expression with N as the exponential constant (Farrow’s constant) and η’ as the viscosity coefficient.

#### 2.3.8. In Vitro Drug Release

A total of 1 g selected formulation was filled into a dialysis tube (Spectrapor, MW cut-off: 6000–8000). The filled dialysis tube was soaked in 100 mL PB at 37 °C with a rotational speed of 50 rpm using a shaking incubator (NB-205, N-Biotek, Bucheon, Korea). Aliquots, with each having a volume of 10 mL, were withdrawn from the release medium at different time intervals and were replaced with 10 mL fresh medium. The VCM release amount was analyzed using ultraviolet (UV)-high-performance liquid chromatography (HPLC) (HPLC 1260 Infinity Series, Agilent Technologies Inc., Santa Clara, CA, USA) equipped with C18 column (150 mm × 4.6 mm, 5 μm particle size). The 0.1% *v/v* phosphoric acid (90%) and acetonitrile (10%) were used as the mobile phase. The injection volume and flow rate were set at 20 μL and 1.5 mL/min, respectively. The eluent was detected by UV-detector at a wavelength of 280 (*n* = 6). The detected values were converted to the amount of VCM by using a standard curve of the drug (*r^2^* = 0.999). The retention time of VCM was 7 min and the limit of detection was 1.1524 μg/mL (from Equation (2)). The crucial mathematical equations, including power-law, zero-order, first-order, and Higuchi’s were used for VCM release profile fitting. The coefficient of determination (*r^2^*) was the parameter for indicating the degree of curve fitting. According to the Korsmeyer–Peppas model, the diffusion exponent (*n*) from the power-law model indicates the drug transport mechanism [43,44,45].
(2)Limit of detection = 3.3 × σS
where σ is the standard deviation of the response and S is the slope of the calibration curve.

#### 2.3.9. Aqueous Phase Influx Tracking

The sodium–fluorescein tracking approach was applied for observation of aqueous phase migration. The 0.003% *w/v* sodium–fluorescein in 0.6% *w/v* agarose was prepared. Sodium–fluorescein was added to the boiled PB, together with the agarose. The obtained solution was poured onto a glass slide. The glass slides were coated with 2 mm-thick sodium–fluorescein-loaded agarose. Then, 100 µL formulation was dropped at the boundary of the settled agarose. The migration of fluorescence color on behalf of the aqueous phase was time-lapse captured under an inverted microscope with 100× magnification (TE-2000U, Nikon, Tokyo, Japan).

#### 2.3.10. Topography of Transformed Matrices under Scanning Electron Microscopy (SEM)

The in situ forming matrix systems after release test were dried overnight at room temperature and kept in a desiccator for one week to avoid the melting and collapse of their structures. The dried samples were coated with gold before being examined by field-emission SEM (Tescan Mira3, Tescan, Brno, Czech Republic) at an accelerating voltage of 15 kV. The porosity of the matrix was determined using image analyzer program (JMicroVision 1.2.7, JMicroVision, Geneva, Switzerland, 2015) [46].

#### 2.3.11. Antimicrobial Activity Studies

The antimicrobial activities of the prepared systems against *S. aureus* ATCC 43300, *S. aureus* DMST 6532, *S. aureus* ATCC 25923, *E. coli* ATCC 8739, and *C. albicans* ATCC 17100 with agar cup diffusion method was determined as previously mentioned [12,41,47]. The 100 µL tested preparation-loaded cylinder was placed on the surface of an inoculated agar plate. The plate was then incubated at 35°C for 18 h. Thereafter, the inhibition zone margin in millimeters (mm) was measured in triplicate. In accordance with the quality control recommendations of Clinical and Laboratory Standards Institute (2017), S. *aureus* ATCC 25923 strain was used as the control species to test the standard antibiotic discs, including cefoxitin (30 g/disc), trimethoprim/sulfamethoxazole (1.25 / 23.75 g/disc), vancomycin (30 g/disc), clindamycin (2 g/disc), and ciprofloxacin (5 g/disc), using the disc diffusion method. For anaerobic bacteria, the test was conducted using an anaerobic incubator (Forma Anaerobic System, Thermo Scientific, Oakwood, OH, USA).

#### 2.3.12. Statistical Analysis

All data were expressed as mean ± standard deviation (SD). Statistical significance was checked by applying one-way analysis of variance (ANOVA) followed by the least significant difference post hoc test and significant differences were indicated when *p* < 0.05.

## 3. Results and Discussion

### 3.1. pH, Density, and Viscosity

Table 2 shows the apparent pH, density, and viscosity of fatty acid solutions. The 35% *w/v* fatty acid solutions showed an increased pH when a higher-MW fatty acid was used due to the less fatty acid molecules. In the case of 1.75 M fatty acid solutions, the decreasing trend of pH was noted. The decrease in pH resulted from the diminishing of solvent given that NMP and DMSO show a higher pH (11.77 ± 0.03 and 11.08 ± 0.38, respectively). However, several fatty acid solutions, such as 1.75CPLN, 1.75CPLD, and 1.75SD showed suspicious results, which possibly arose due to the following reasons. The hydrophobicity of long hydrocarbon chains can dominate the acidity of the carboxyl group, and long-chain fatty acids have high pK values [48,49], which results in the low protonation of fatty acids [50]. By contrast, the lower carbon amount of CPL showed a less dominating effect on the carboxyl group, with its ionization resulting in the significantly lower pH than the others.

The descending density was distinguished once the concentration of fatty acid was fixed at 1.75 M. The density of the mixed compound was the sum of each component density based on their mass fraction [51]. The lowering trend of density was due to the accumulation of fatty acids, which are low-density material. Similarly, the fatty acid solutions using DMSO as a solvent produced a high density due to the high density of this solvent. The densities of NMP, DMSO [52], CPR, LAU, MYR, PAL, and STR were 1.027, 1.100, 0.85 ± 0.08, 0.82 ± 0.08, 0.86 ± 0.09, 0.9 ± 0.09, and 0.84 ± 0.08 g/cm^−3^, respectively, at 23–24 °C [53]. Moreover, according to the viscosity results, the trend of viscosity change was similar to that of density. It was reported that these two values are related [54].

All the solutions had low viscosity given that a simple and small structure of fatty acid can lessen the inter/intramolecular interaction, different from bulky molecules such as polymer. Eudragit^®^ RS, a gel former of recently developed in situ forming gel, can readily interact with vehicles, resulting in a steep viscous preparation when the concentration is increased [55]. Comparing between the formulations, a higher viscosity was associated with the use of a higher MW fatty acid. In the study of binary liquid mixtures of fatty acids, the solutions exhibited an increased viscosity when the ratio of long-chain fatty acid was increased [56]. The following reason might explain the increase of the viscosity. Firstly, the interaction from carboxyl group via hydrogen bonding. The functional group as carboxyl was perceived for behavior of fatty acids in solutions. It was reported that the carboxyl group of fatty acid protruding from the cavity of amylose-lipid complexes can interact with other molecules, resulting in increased viscosity [57,58]. Although the hydrogen bonding between carboxylic groups can typically promote viscosity [59], the hydrophobic interaction should be decisively considered. The hydrophobic interaction can enhance viscosity [60]; the long chain length of alkanes induces the increase in hydrophobic interaction, resulting in the high viscosity of silicone oil [61]. However, the results from the 35% *w/v* fatty acid series might not be distinct since the smaller number of fatty acid molecules were evident for high MW fatty acid-based solutions.

### 3.2. In Vitro Matrix Formation

The results showed the changing state behavior from the liquid state into the opaque solid-like state of fatty acid solutions by time after exposure to an aqueous environment by the antisolvent process. The use of different MW fatty acids remarkably affected the in vitro apparent opaque matrix formation behavior (Figure 1). The prerequisite of in situ forming matrix is the capability to form into a solid-like matrix within the expected time. Within 5 min, all formulations comprising CPL and CPR could not transform into a solid-like state; the same was observed for LAU in NMP. Meanwhile, rapid formation was acquired apparently, if the high-MW fatty acids, including MYR, PAL, and STR, were used. LAU, a 12-carbon atom fatty acid, showed a conflicting formation behavior. LAUN matrix could not be formed, although LAUD could. LAU located itself amid the solvent effect. Notwithstanding, the inquiry of completed formation or the rate of formation could not be achieved in this experiment. Thus, the determining matrix formation at the cross-sectional view was performed.

### 3.3. Matrix Formation Rate (Cross-Sectional View)

The transformation behavior of the formulations in the agarose corresponded well with the result of a previous experiment (Section 3.2). The low-MW fatty acid solutions containing CPL and CPR did not change into a solid-like state (Figure 2A). The trend of diminishing transformation rate by time (Figure 2B) resulted from the restriction of solvent transfusion through interfacial network of initiated fatty acid matrix, which became denser with higher tortuosity and lower porosity by time. Similarly, the dense gelation network also lowered the water diffusion coefficient [62]. Therefore, several formulations, including 35LD, 1.75LD, 35MD, and 1.75MD, showed incomplete matrix formation within 15 min. Crystallization was another crucial factor. Although the aqueous/solvent diffusion was obstructed, the crystal growth of the fatty acid still occurred continuously. Likewise, the completed PAL and STR matrix rapidly formed. The thermodynamic stability could be interrupted by various factors, such as the presence of new interfacial energy from the initial crystal, aqueous surface, bubble surface, etc., together with the supersaturation (during decreasing of solvent); afterward, the system stabilizes itself, resulting in crystal formation [63]. In this experiment, the difference in matrix formation between systems using NMP and DMSO as solvent was noticed. At the 1–5 min interval, the formulations using NMP as a solvent did not transform, whereas the formulations using DMSO showed transformation visibly at the boundary. This behavior was noted in all fatty acid solutions in addition to comparison with high-MW fatty acids such as PAL. The DMSO diffusion rate was higher than that of NMP [7] due to the lower viscosity of DMSO. However, the fundamental properties, such as hydrophilicity/lipophilicity, should not be disregarded, especially in the in situ forming matrix, which involves hydrophobic materials and an aqueous environment. Owing to its higher hydrophilicity (logK_ow_ values of DMSO and NMP are −1.98 and −0.38, respectively [64,65]), DMSO can mix with high polarity water more easily, whereas NMP remained still. Otherwise, NMP had a higher carrying capability for fatty acids although the system was mixed with inward water, as mentioned and disclosed in Section 3.6.

### 3.4. Interfacial Phenomenon during Initial Experimental Period

This experiment was set up to simulate the phenomenon at the boundary of formulation, which faced the target site containing the biological fluid. The agarose platform referred to the simulated biological site for which the in situ forming systems were administered and transformed into solid-like material to modulate drug release as mentioned previously. Comparison between solvents using DMSO (Figure 3) caused an outright liquid movement, whereas the delayed fluid movement was evident for all formulations using NMP as a solvent. At the early initial experimental period, these events confirmed the previously mentioned phenomenon indicating that DMSO more readily mixed with the aqueous phase than NMP. Although the liquid movement in NMP-used formulations was slightly noticed, the transformation continued. The faster rim crystal formation of 35PN compared with 35MN indicates that using higher MW fatty acid promoted a more rapid phase inversion with antisolvent process. Considering the results in Section 3.2 and Section 3.3, the MW and solvent type played considerable roles in matrix formation. The type of solvent influenced intensely the matrix formation of LAU-loaded solution. The rapid crystallization dominated massively in the long-chain fatty acid solution. Theoretically, an energy exists at the boundary between any unmixable objects; this energy can provoke nucleation and subsequently, crystallization through a thermodynamic driving force [66]. Thence, not only the external interface comprising agarose and initial crystal interface but also the interface of unmixed liquids can induce continuous crystallization. The additional investigation is explained in Section 3.5.

### 3.5. Electrical Potential Difference during Phase Transformation

When the systems meet the aqueous environment, the solvent diffuses outward as the water diffuses inward, leading to the in situ formation with antisolvent process, that is, the water ratio increases over time. For this reason, in this experiment, the water amount (% *v/v*) was implied to correspond with the time of phase transformation. The interfacial tension is related to nucleation [67] and the relation between electrical potential and interfacial tension was pointed out [68,69]; therefore, the electrical potential difference was applied to track the nucleation process. The increase in electrical potential values (Figure 4A) is related to the enriched high polarity phase owing to the high amount of water and the ionization of fatty acid molecules [70]. When the polarity of the system reached the point at which the hydrophobic part could not be retained, the two liquids separated, resulting in a new water solvent/lipid solvent interface. The appearance of the interface before nucleation was detected (Figure 4B). It was called a “dense liquid” in another study [71]. The boundary between the two liquids gained a free energy; as a result, they separated to reduce the energy via nucleation and crystallization [70]. Lastly, at the point of phase separation, the value steeply decreased owing to the absence of a polar group as carboxyl because of the crystallization into solid fatty acid. By comparison, the separation point of long-chain fatty acids was obtained at a low electrical potential difference. The dissolved long-chain fatty acids might have shown less tolerance during electrical change given that they have a lower polarity than short-chain fatty acids. This result corresponded to that of another report, which indicated that a smaller surface charge contributes to the water interface of long-chain fatty acids [72]. Given the relation of interfacial tension and electrical potential, this experiment assumed that the interface of two liquids, that is, water solvent/lipid solvent, was generated via a difference in voltage value. This assumption also means that an electrical property can be one of the factors that induce the phase separation of fatty acid-based in situ forming systems.

### 3.6. Influence of KH_2_PO_4_ on Water Tolerance

KH_2_PO_4_ was used to prepare buffer of various pH and a simulated physiological fluid [7,55], and ions affect the separation of hydrophobic material [73]. Therefore, the effect of KH_2_PO_4_ on water tolerance of fatty acid solutions was evaluated. The high concentration of KH_2_PO_4_ notably lowered the separation point of all formulations (Figure 5A). The ionic kosmotropes of HPO_4_^2^^−^ promoted the high stability and proper structure of water–water interactions; therefore, this electrolyte forced the phase separation of hydrophobic substances [73]. In other words, kosmotrope promoted the interfacial tension [69]; thus, the separation point of these fatty acids occurred earlier when the concentration of KH_2_PO_4_ was increased. With the increased hydrophobicity and low aqueous solubility of long-chain saturated fatty acids, a supersaturated point was easily obtained. Hence, the component of the aqueous environment should be considered along with the properties of the formulation to govern the antisolvent-induced in situ formation process of fatty acid.

### 3.7. Injectability

The injection force values of fatty acid formulations were in the range of 1.55–3.12 N, which are significantly lower than acceptable criteria (<50 N [74]), indicating the high injectability, with no significant difference between each formulation (*p* < 0.05). The injection work values of fatty acid-based formulation were in the range of 16.9103–35.3128 N.mm (Figure 5B), which is related to their MW and corresponds to their density and viscosity. Another report suggests that this property might be due to the lubricating performance of fats [75]. This high injectability of the fatty acid-based in situ forming system indicates the ease of administration via injection. For example, in the case of knee joint administration, this system is more easily injected through a small needle (27-G) than that of other intra-articular dosage forms and hyaluronic acid solutions, such as Atri-III^®^ and Hyalgan^®^, which are highly viscous fluids [76] where the cross-linked hyaluronic acid molecules with divinyl sulfone and hyaluronic acid solution exhibited 12.5 and 35 N through a 22-G needle [77]. Moreover, all of the formulations in fatty acid-based series exhibited Newtonian flow behavior in which the flow index (N) was not statistic different. Even for the lowest injectability 1.75SN, a Newtonian behavior was obtained (N = 1.000 ± 0.006). This flow behavior was an expected pattern in an injection dosage form [78].

### 3.8. In Vitro Drug Release

At the initial experimental periods, the VCPLD formula showed the highest burst release because it could not form a solid-like state. The burst release of a well-transformed matrix was influenced by the type of fatty acid. VPD, VMD, VCPRD, and VLD presented a descending burst release (Figure 6). Although, the crystal network at the contacted surface was rapidly formed, as described in Section 3.3 and Section 3.4. VMD and VPD exhibited more burst release than VLD. The rapid matrix formation signified the rapid movement of DMSO with the crystallization of fatty acid. Therefore, the rapid DMSO outflux carried this drug with its diffusion. Moreover, the evidently large passage of high fatty acid-based formulation might be formed. The topography of the obtained matrix under SEM revealed this evidence. According to the results in Section 3.6, 35LD had a higher separation point, signifying a lower degree of supersaturation than 35MD and 35PD. Thus, VLD needed more water penetration to induce phase separation, resulting in a high water-containing volume at the given time. Then, the high number of fatty acid nuclei coincided with the slowed crystal growth rate of each other [79]. Each crystallization origin grew with vignettes, resulting in less porosity and high tortuosity. By contrast, the rapid formation had less crystallization origin. Thus, the large passage mentioned above can be obtained via continuous crystalized growth. Thereafter, with a completed matrix formation, the drug liberation attained as a sustaining release pattern, whereas the control group showed a fast and complete VCM release within the first day. VMD and VPD prolonged the drug release over seven days, with about 80% cumulative drug release, whereas VLD showed 56% cumulative drug release. The formulations gained sustainable capability through the restriction of water and drug migration [62,80]. The hydrophobicity of fatty acid and the obtained hard matrix retarded the water–drug movement. This movement retardation was also observed when using the other hydrophobic matrix formers, such as bleached shellac and cholesterol [13,80]. Hydrogen bonding was also considered given that fatty acid and VCM molecules contain the functional groups COOH, OH, and NH. Although the formulations sustained the drug release over seven days with a 56–80% drug release, repeated administration can only be conducted if the treatment period is extended for a longer period of time. The estimated *r^2^* from the VCM cumulative release data fitted to different mathematical release models is shown in Table 3. All release profiles fitted well with Higuchi’s equation, which depicted the migration of VCM through the tortuosity matrix in which the distance from the matrix surface to the drug dissolution region was accounted [81,82]. For the nonswelling system with a cylindrical shape, the diffusion exponent (n) implies the transport of drug, where *n* = 0.89 indicates the case II transport mechanism, *n* = 0.45 indicates Fickian diffusion, and 0.45 < *n* < 0.89 indicates anomalous diffusion. The Fickian diffusion refers to the drug transport mechanism in which the diffusion rate is faster than that of structure relaxation. Whereas, when the relaxation process is higher than diffusion, it is Case II transport. Otherwise, when these rates are close to each other, anomalous diffusion is acquired [44,45,83,84]. According to those cited, the VCM was released from VLAUD and VMYRD via the mechanism of anomalous diffusion while VPALD presented a pure Fickian diffusion.

### 3.9. Aqueous Phase Influx Tracking

Although the solvent exchange was mentioned in many reports [7,79] and the solvent tracking inquiry was done [55], the movement of the aqueous phase has not been profoundly understood. Thus, in this study, the aqueous influx tracking of the following formulations was investigated (Figure 7). The sodium fluorescence color represented the aqueous phase from the agarose stage, which gradually moves inward. VCPLD and VCPRD, which could not form a matrix, presented a rapid aqueous influx at the beginning of the experiment. VLD showed a classical pattern. Firstly, the aqueous influx diffused inward; secondly, the surface network was formed, and aqueous diffusion was retarded consequently. For the transformation of VMD and VPD, the crystal edge near the aqueous front indicated the high transformation capability of these long-chain fatty acids. VMD showed a faster crystal formation along with aqueous diffusion than VPD, which confirmed the rapid dense network of high-MW fatty acids such as PAL and the retardation of aqueous-solvent migration. Although the solvent exchange was prohibited via dense network of PAL at the initial experimental period, the notable crystallization capability of PAL appeared and accomplished the completed matrix formation, as described in Section 3.3. Generally, if a high-MW fatty acid (PAL) forms a rapid dense network and gradually retards the solvent exchange process, the transformation from a solution into a solid would be prohibited owing to the restriction of water diffusion. However, this phenomenon was not observed. Once the first crystal was formed, the crystal growth initiated its self-transformation capability, which was significantly more dominating than solvent exchange, as described above and in Section 3.3. These results confirm that solvent exchange and crystallization capability are crucial parameters influencing the self-formation process of the fatty acid-based in situ forming matrix. MYR-based formulation showed a well-balanced effect on solvent exchange and crystallization. By contrast, VLD, which had high free migration of solvent at the initial time of the experiment, exhibited less crystallization capability with a higher number of nuclei formation; once these nuclei grew, a very dense network was obtained and the burst drug release was minimized subsequently, as mentioned in Section 3.8. Therefore, solvent exchange and crystallization should be managed and considered to design further antisolvent-induced in situ forming matrix systems.

### 3.10. Topography of Transformed Matrices

On behalf of low-MW fatty acid-based formulation, the dried matrix after the release test of VLD was selected to compare with the higher one as VPD (Figure 8). The VLD dried matrix presented a needle-shaped crystal agglomerate with a continuously intricated matrix (average porous size = 4.205 μm). However, VPD showed an enclosed sheath crystal with a large and simple aperture (average porous size = 6.376 μm). The excavation of VPD confirmed the low tortuosity that resulted in a burst drug release and low sustainable drug liberation as mentioned in Section 3.8. The continuously intricated matrix of VLD prolonged the release of VCM owing to its high tortuosity and lower porosity, whereas the pattern of VLD crystal formation was related to the following step. Firstly, the water diffused inward in a high unit per volume, which sufficiently generated a large number of nuclei per volume simultaneously. Secondly, the nuclei growth individually showed with a competition to each other, resulting in a slow crystal growth. Thirdly, the tangled structure of the matrix was formed via a lot of crystallization origin. By contrast, according to the above results, VPD was ready to generate a crystal with a high crystal growth capability and that the nuclei were formed rapidly at the interface of water/formulation. When the initial nuclei were formed, a rapid crystal growth occurred consequently with no competition effect because it contained less nuclei per volume. Thus, VPD had a lower crystal origin, which resulted in a matrix topography with high porosity and low tortuosity, in which the drug leaked easily.

### 3.11. Antimicrobial Activity Studies

Table 4 presents the antibacterial activities for drug-loaded and drug-free formulations. The drug-loaded formulation exhibited a significantly higher antibacterial activity against *S. aureus* DMST 6532, *S. aureus* ATCC 25923, *E. coli* ATCC 8739, and MRSA strains (*S. aureus* ATCC 43300) than the drug-free formulations (*p* < 0.05). No statistical difference (*p* < 0.05) was observed between the antimicrobial activity of drug-loaded and drug-free formulation against *C. albicans,* which indicates that the drug showed no effect on *C. albicans*. Thus, the activity against *C. albicans* was due to the short-chain fatty acids. Short-chain fatty acids exert activity against *C. albicans* by inhibiting germ tube and hyphae formation and reducing the metabolic activity [85]. The descending trend of antimicrobial activity was related to the use of higher MW fatty acids, which retarded the VCM release. Notably, the activity of VMD and VPD against MRSA was significantly decreased compared with that of VD. Although this activity of most drug-loaded formulations was no different from that of VD, the following formulations provoked a significantly high activity (*p* < 0.05), indicating a synergistic effect: VLD (against *S. aureus* DMST 6532); VCPLD; and VCPRD (against *S. aureus* ATCC 43300 and *S. aureus* ATCC 25923). The antimicrobial activity of short-chain fatty acid has been determined through various mechanisms, such as inhibiting the entry of essential molecules into bacterial cells [86], interrupting membrane fluidity, and disrupting catalytic activity [87,88]. Several microbes are highly sensitive, resulting in the synergistic effect. This section’s results depicted not only the relation between antimicrobial activity and fatty acid type as mentioned but also the formed matrix structure. As seen in the previous section, the structure of VLD and VPD matrices are distinct. The higher porosity, smaller pieces, and needle shape crystals of VLD formulation might relate to the higher antimicrobial activity. The relation between these characters and antimicrobial activity was also mentioned in other studies where the high porous and small particle size could enhance antimicrobial activity [89,90].

Therefore, the use of a VCM-loaded fatty acid-based in situ forming matrix not only modulated the drug release capability but also effectively inhibited various pathogens, especially MRSA. However, a significant difference was noted in multiple properties that rely on the MW of fatty acid. Hence, to design a proper fatty acid-based in situ forming matrix for any aspect of drug delivery applications, the MW of this matrix former should be considered. In order to clinically translate, the limit of DMSO and pharmacokinetic of VCM depending on an application site should be considered. DMSO has been utilized as a medium of many dosage forms including in situ forming implants [11,91], in situ forming microparticle [92], and intra-arthricular injection [93]. 50% *w/w* DMSO (RIMSO-50^®^) is approved by US FDA for treatment of human interstitial/chronic cystitis as well as various topical dosage form containing DMSO for flap ischemia (60% DMSO), herpes zoster (5% idoxuridine in DMSO), and injection site extravasation (99% DMSO) treatment [94]. Whilst the safety data with various medical applications of this solvent indicate the probability of using DMSO in the developed dosage form, the formulation such as the developed VLD, VMD, and VPD need to be determined more in a clinical experiment. For example, in the case of postoperative knee infection, the 1M DMSO did not provoke toxicity to the chondrocytes from the cell recovery with toxicity kinetic study [95]. The estimate DMSO concentration after injection of 0.5–3.0 mL of developed formulation into 5–30 mL synovial fluid was in the rank of 0.004–0.25 M and the estimated concentration of VCM were above MRSA MIC level over 6 days, based on 3.22 h^−1^ VCM half-life in synovial fluid and 2–4 μg/mL MRSA MIC level [33,96].

## 4. Conclusions

Fatty acids are considered beneficial materials because of their biocompatibility, biodegradability, and versatile usefulness. The series of fatty acid-based in situ forming matrix showed markedly low viscosity and apparent high injectability promoting their ease for injection. The type of fatty acids affected the transformation behaviors at both microscopic and macroscopic levels. Their transformation rates have relied on MW and solvent characters. The VLD, VMD, and VPD prolonged the release of VCM over seven days due to their hydrophobicity and high matrix tortuosity. The release pattern of VLD was different from that of VMD and VPD due to the nucleation/crystallization and solvent exchange behaviors. The VCM-loaded fatty acid in situ forming matrices via antisolvent process showed efficient antimicrobial activities against various pathogens including MRSA. Hence, the developed preparations, namely, VLD, VMD, and VPD, can become suitable local antimicrobial delivery systems for various Gram-positive bacterial infection sites, such as joints, crevicular pockets, and postsurgery sites. The administration amount can be adjusted based on the following considerations: the fluid volume at the target site, the minimum inhibition concentration of any bacteria at the infection site, and the pharmacokinetics of VCM at the target site. However, further investigation for the safety and clinical efficacy of a particular infection site should be investigated. This study provided fundamental information on further development using various fatty acids as a matrix former for antisolvent-induced in situ forming system.

## Figures and Tables

**Figure 1 pharmaceutics-12-00808-f001:**
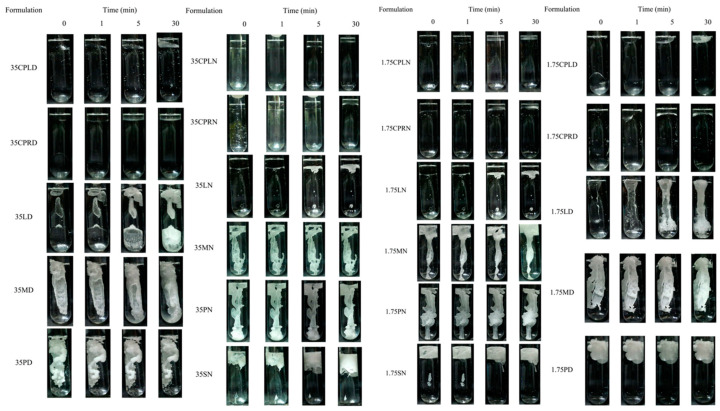
In vitro matrix formation behavior of fatty acid-based in situ forming matrix in PB with pH 7.4 (triplicate testing; *n* = 3).

**Figure 2 pharmaceutics-12-00808-f002:**
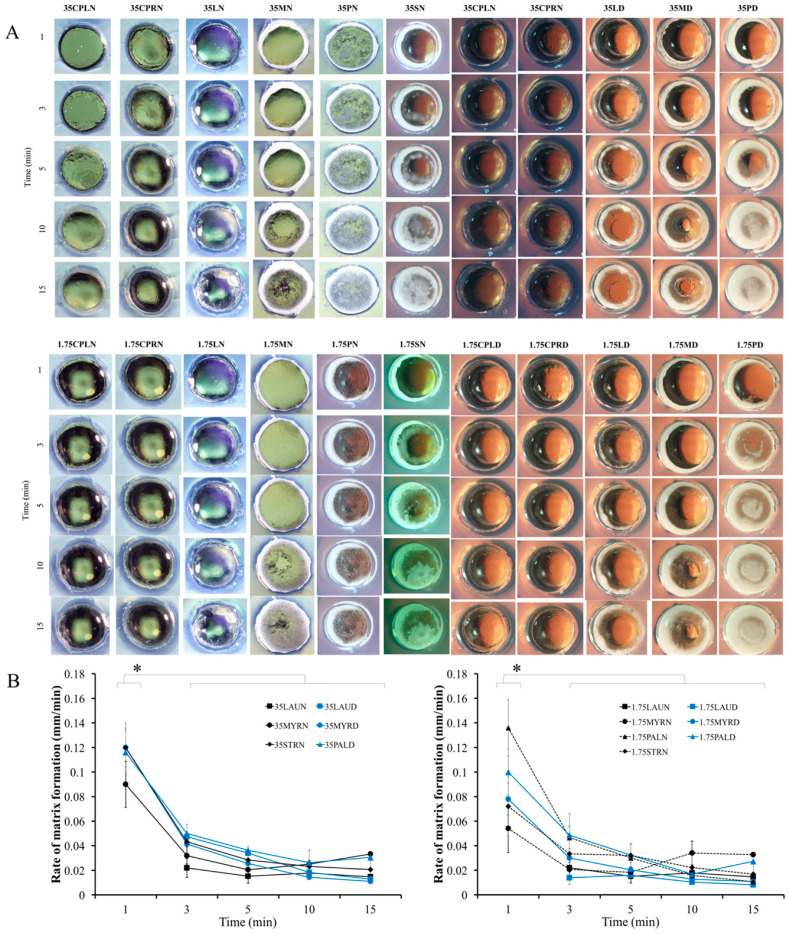
Cross-sectional view of matrix formation (**A**) and its rate over time in the agarose well (**B**) (triplicate testing; *n* = 3). * represents *p* < 0.05.

**Figure 3 pharmaceutics-12-00808-f003:**
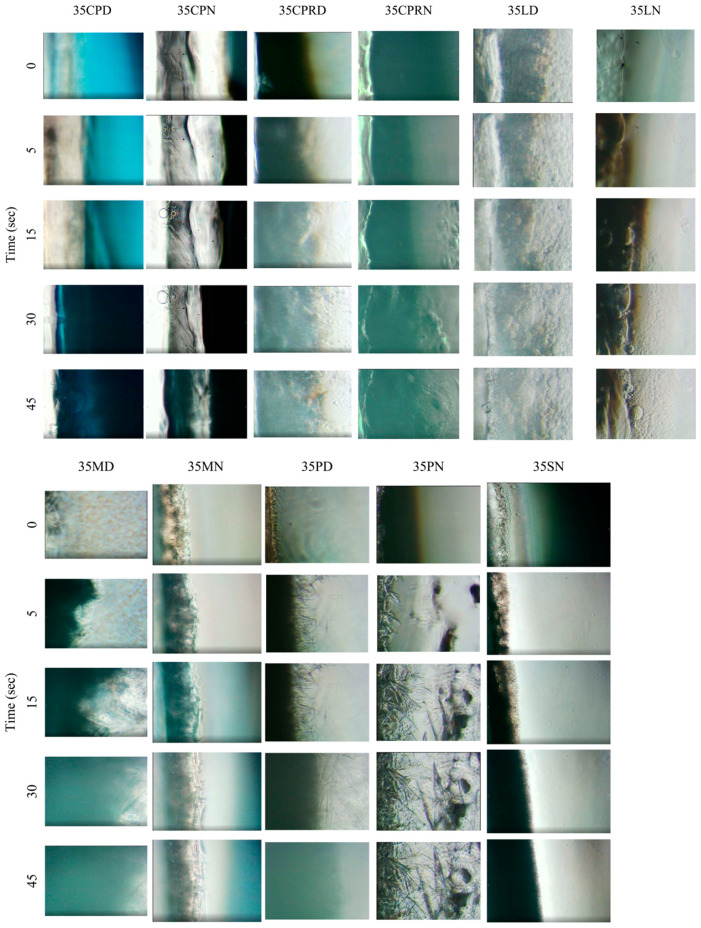
Behaviors of preparations at the boundary of aqueous phase and in situ forming systems during the initial period (0–45 s) at 100− magnification under inverted microscope.

**Figure 4 pharmaceutics-12-00808-f004:**
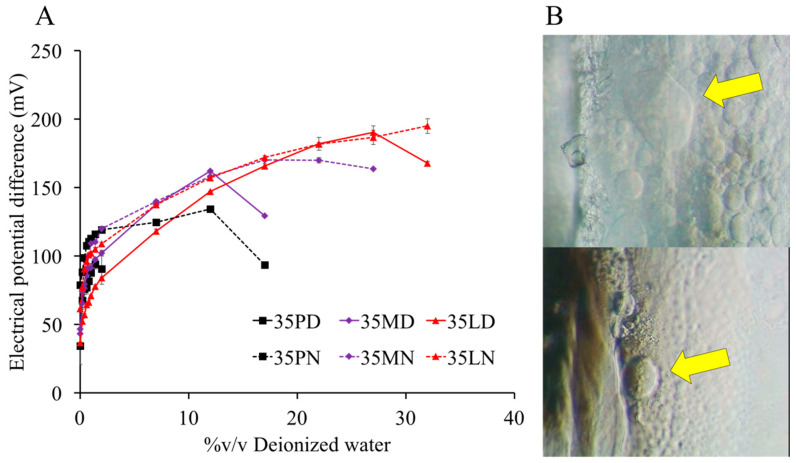
Electrical potential difference of systems (preparation and water) during in situ formation process (**A**) (triplicate testing; *n* = 3). The appearance of dense liquid at 45th s of 35LD (top) and 35LN (beneath) (**B**) at 400− magnification under inverted microscope.

**Figure 5 pharmaceutics-12-00808-f005:**
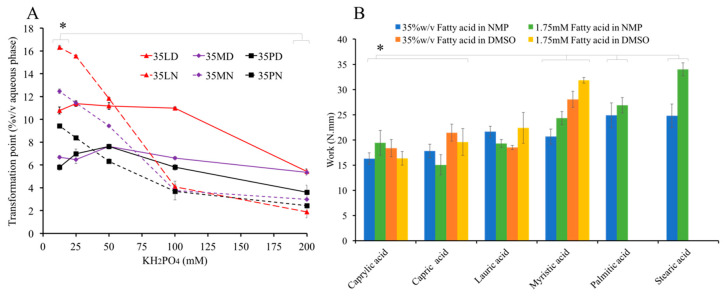
Effect of KH_2_PO_4_ concentration on transformation point (**A**). Injectability of preparations (work) (**B**) (triplicate testing; *n* = 3). * represents *p* < 0.05.

**Figure 6 pharmaceutics-12-00808-f006:**
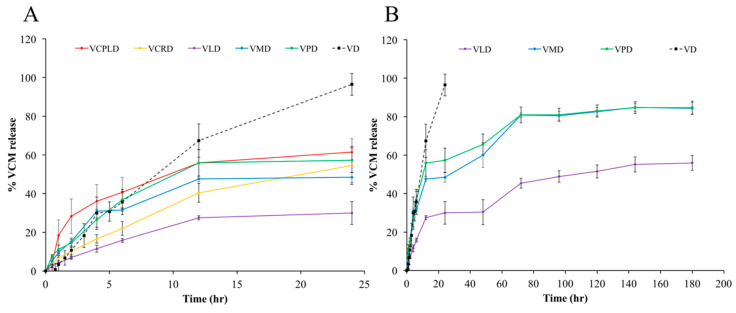
VCM release from in situ forming matrix at the initial experimental period (**A**) and over 7 days (**B**) in PB at pH 7.4 using dialysis tube method.

**Figure 7 pharmaceutics-12-00808-f007:**
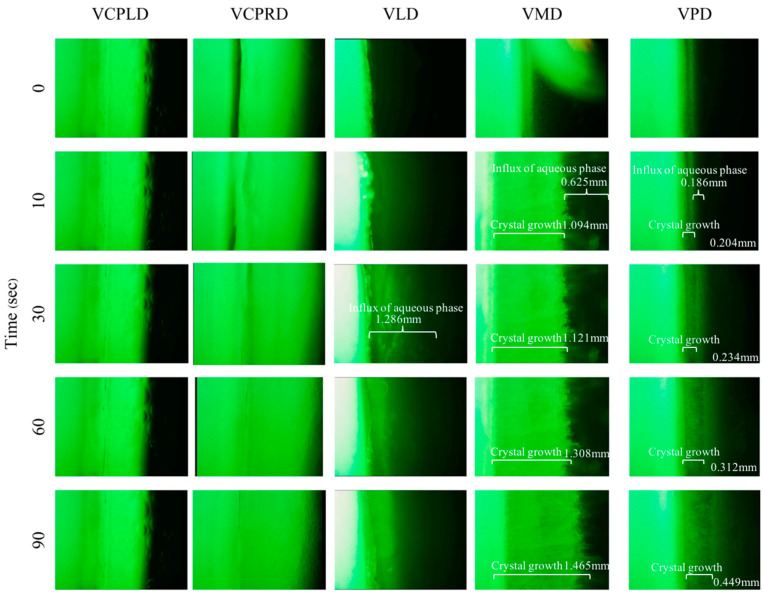
Influx behavior of aqueous phase at boundary of drug-loaded fatty acid in situ forming matrix at 400− magnification under inverted microscope.

**Figure 8 pharmaceutics-12-00808-f008:**
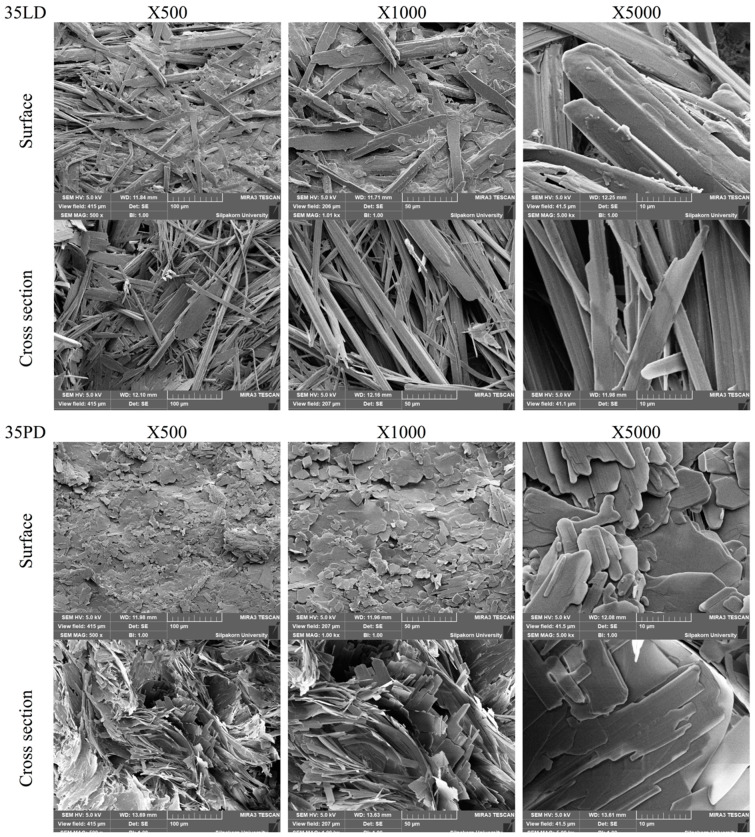
Topography under scanning electron microscope of formed matrix after release test.

**Table 1 pharmaceutics-12-00808-t001:** Composition of various drug-free in situ forming matrix systems.

Formulation Code	Fatty Acid		VCM (1 g)	Solvent
	Type	Amount (g)		(Adjust to 100 mL)
35CPLD	CPL	35.06	-	DMSO
35CPRD	CPR	35.06	-	DMSO
35LD	LAU	35.06	-	DMSO
35MD	MYR	35.06	-	DMSO
35PD	PAL	35.06	-	DMSO
35SD	STR	35.06	-	DMSO
1.75CPLD	CPL	25.24	-	DMSO
1.75CPRD	CPR	30.15	-	DMSO
1.75LD	LAU	35.06	-	DMSO
1.75MD	MYR	39.97	-	DMSO
1.75PD	PAL	44.88	-	DMSO
1.75SD	STR	49.78	-	DMSO
35CPLN	CPL	35.06	-	NMP
35CPRN	CPR	35.06	-	NMP
35LN	LAU	35.06	-	NMP
35MN	MYR	35.06	-	NMP
35PN	PAL	35.06	-	NMP
35SN	STR	35.06	-	NMP
1.75CPLN	CPL	25.24	-	NMP
1.75CPRN	CPR	30.15	-	NMP
1.75LN	LAU	35.06	-	NMP
1.75MN	MYR	39.97	-	NMP
1.75PN	PAL	44.88	-	NMP
1.75SN	STR	49.78	-	NMP
VD	-	-	1.00	DMSO
VCPLD	CPL	35.06	1.00	DMSO
VCPRD	CPR	35.06	1.00	DMSO
VLD	LAU	35.06	1.00	DMSO
VMD	MYR	35.06	1.00	DMSO
VPD	PAL	35.06	1.00	DMSO

CPL: Caprylic acid, CPR: Capric acid, LAU: Lauric acid, MYR: Myristic acid, PAL: Palmitic acid, STR: Stearic acid, DMSO: Dimethyl sulfoxide, NMP: *N*-methyl-2-pyrrolidone, VCM: Vancomycin HCL.

**Table 2 pharmaceutics-12-00808-t002:** Physical properties of fatty acid-based in situ forming matrix (triplicate testing; *n* = 3).

Formulation	pH	Density (g/cm^−3^)	Viscosity (cPs)
35CPLD	5.55 ± 0.07	1.0240 ± 0.0002	6.29 ± 0.04
35CPRD	6.13 ± 0.02	1.0205 ± 0.0001	7.51 ± 0.02
35LD	6.05 ± 0.11	1.0161 ± 0.0002	8.05 ± 0.05
35MD	6.19 ± 0.11	1.0142 ± 0.0000	9.54 ± 0.17
35PD	6.22 ± 0.05	1.0119 ± 0.0001	10.90 ± 0.05
35SD	6.78 ± 0.01	1.0109 ± 0.0001	ND
1.75CPLD	5.97 ± 0.02	1.0407 ± 0.0001	5.47 ± 0.12
1.75CPRD	6.44 ± 0.04	1.0345 ± 0.0001	6.08 ± 0.05
1.75LD	6.15 ± 0.03	1.0154 ± 0.0001	7.95 ± 0.10
1.75MD	5.95 ± 0.03	1.0026 ± 0.0002	11.86 ± 0.08
1.75PD	5.98 ± 0.04	0.9899 ± 0.0002	ND
1.75SD	6.28 ± 0.01	0.9799 ± 0.0002	ND
35CPLN	5.35 ± 0.01	0.9894 ± 0.0001	7.07 ± 0.20
35CPRN	5.76 ± 0.05	0.9825 ± 0.0002	7.96 ± 0.29
35LN	5.93 ± 0.03	0.9797 ± 0.0001	8.43 ± 0.09
35MN	5.98 ± 0.01	0.9767 ± 0.0001	8.91 ± 0.40
35PN	6.06 ± 0.01	0.9731 ± 0.0001	8.92 ± 0.54
35SN	6.00 ± 0.00	0.9759 ± 0.0001	9.24 ± 0.44
1.75CPLN	6.53 ± 0.05	0.9935 ± 0.0001	5.63 ± 0.06
1.75CPRN	6.53 ± 0.05	0.9947 ± 0.0000	5.73 ± 0.08
1.75LN	5.81 ± 0.04	0.9775 ± 0.0000	7.57 ± 0.16
1.75MN	5.60 ± 0.01	0.9664 ± 0.0001	9.49 ± 0.20
1.75PN	5.41 ± 0.05	0.9579 ± 0.0001	10.90 ± 0.20
1.75SN	5.16 ± 0.01	0.9529 ± 0.0003	13.97 ± 0.05

ND: not determined; by comparison in each group of 35% and 1.75 M fatty acids in different solvents, the different MW of fatty acid resulted in a significantly increased/decreased trend (ANOVA, *p* < 0.05). According to a post hoc test, any paired samples showed significant differences (*p* < 0.05), except the following: pH of 1.75MD–1.75PD; pH of 35CPRD–35LD–35MD–35PD; viscosity of 1.75CPLN–1.75CPRN; and viscosity of 35LD–35MD–35PD–35SD.

**Table 3 pharmaceutics-12-00808-t003:** Degrees of goodness-of-fit and estimated parameters from curve fittings of the release profiles of vancomycin HCl (VCM)-loaded in situ forming systems in phosphate buffer at pH 7.4 using the membrane-less method.

Formula	Zero-Order		First-Order		Higuchi’s		Korsmeyer–Peppas		
	*r^2^*	*k*	*r^2^*	*k*	*r^2^*	*k*	*r^2^*	*k*	*n*
VLD	0.8374	0.3113	0.5863	0.0056	0.9584	4.4057	0.9593	0.0498	0.5062
VMD	0.8130	0.7792	0.5470	0.0088	0.9451	8.3685	0.9334	0.1048	0.4923
VPD	0.7537	0.7772	0.5576	0.0082	0.9169	8.5386	0.9471	0.1239	0.4589

*r^2^* = coefficient of determination, *k* = constant, *n* = diffusion exponent.

**Table 4 pharmaceutics-12-00808-t004:** Inhibition zone diameter of drug-free formulation, drug-loaded formulation, drug in DMSO and DMSO against *S. aureus, E. coli*, and *C. albicans* (triplicate testing; *n* = 3). (- : no inhibition zone).

Formula	Clear Zone Diameter (mean ± SD)				
	*S. aureus* ATCC 25923	*S. aureus* ATCC 43300	*S. aureus* DMST 6532	*E. coli* ATCC 8739	*C. albicans* *ATCC 17100*
35CPLD	11.00 ± 0.00	10.00 ± 0.00	10.00 ± 0.00	10.33 ± 0.58	40.33 ± 2.52
35CPRD	7.67 ± 0.58	7.00 ± 0.00	7.00 ± 0.00	-	24.33 ± 0.58
35LD	7.33 ± 0.58	8.33 ± 2.31	10.00 ± 0.00	-	16.33 ± 0.58
35MD	7.00 ± 0.00	7.00 ± 0.00	7.00 ± 0.00	-	16.67 ± 1.53
35PD	7.00 ± 0.00	7.00 ± 0.00	7.00 ± 0.00	-	13.67 ± 1.53
VCPLD	30.33 ± 1.53	30.00 ± 0.00	27.33 ± 2.08	17.33 ± 0.58	39.33 ± 1.15
VCPRD	31.67 ± 2.08	29.67 ± 0.58	28.00 ± 2.00	16.33 ± 0.58	28.00 ± 2.00
VLD	28.00 ± 3.61	28.00 ± 1.00	34.67 ± 3.06	16.67 ± 0.58	16.67 ± 0.58
VMD	27.00 ± 1.73	27.00 ± 0.00	31.67 ± 2.08	15.33 ± 1.53	15.33 ± 1.15
VPD	26.33 ± 1.53	27.67 ± 1.53	26.67 ± 0.58	13.00 ± 1.00	14.33 ± 0.58
VD	27.33 ± 0.58	29.67 ± 0.58	28.33 ± 1.15	16.67 ± 0.58	18.33 ± 0.58
DMSO	7.67 ± 1.15	6.67 ± 0.58	7.00 ± 0.00	13.00 ± 1.00	19.67 ± 0.58

Attributed to CLSI 2017 for validation; quality control ranges for S. aureus ATCC 25923: cefoxitin, 23–29 mm; trimethoprim–sulphamethoxazole, 24–32 mm; vancomycin 17–21 mm, clindamycin 24–30 mm; ciprofloxacin, 22–30 mm. S. aureus ATCC 25923 could be clearly inhibited by cefoxitine (28 mm), trimethoprim–sulphamethoxazole (28 mm), vancomycin (17 mm), clindamycin (25 mm), and ciprofloxacin (29 mm) discs within the acceptable range of diameter.

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
