# Peer review of "Saturated Fatty Acid-Based In Situ Forming Matrices for Localized Antimicrobial Delivery"

_pharmaceutics, 2020, doi:10.3390/pharmaceutics12090808_

Round 1
Reviewer 1 Report
Article describes saturated fatty acid based DMSO or NMP in situ forming vancomycin loaded formations towards an antimicrobial in situ gel forming liquid.
Please see my comments below:
- Title can be clearer to read
- Abstract can be re-written to provide more quantitative information
- English needs to be carefully revised throughout the manuscript.
- Figure 1 is not providing really any benefit, please remove.
- “Vancomycin HCl (VCM), a bulky hydrophilic drug (Fig. 1G), has been widely used to treat infections caused by Gram-positive bacteria, especially Staphylococcus aureus (including MRSA), S. epidermidis, Streptococcus spp., viridans streptococci and Enterococcus spp [28-31]”. Please state Ic50 against these species for vancomycin with appropriate references.
- VCM is the drug of choice for MRSA infection. – Please provide a reference to support this statement.
- Please state purity of all lipids, solvents and materials.
- How was the pH of the formulation evaluated in a non-aqueous solvent?
- For conductomer experiments, please state values obtained for water as a standard.
- Please state retention time obtain using HPLC method and lower limit of detection.
- The 35% w/v fatty 218 acid solutions showed an increased pH when a high-MW fatty acid was used due to the decrease in fatty acid molecules with a carboxyl group. Shouldn’t the opposite be true? The higher the mw of fatty acid, the lower the moleculers, thus lower content of carboxylic acids?
- I am not really certain about validity of pH measurements within pure solvent mixtures? What does that really show?
- Lines 236-253 are unclear. Cannot follow clearly the arguments made and are confused. Please re-write.
- Table 2 correct Dengityg
- Statistics for Figure 3 are missing
- Little evidence of crystallinity is provided although discussed. Not sure how provided data support this discussion.
- Is unionized water – deionized? Figure 5
- Figure 6 a and b – statistics are missing
- Figure 8 – Is quantification possible using Image J?
- Figure 9- Calculate porosity using Image J
- Would this in situ gel be able to be clinically translated and if so which application is envisaged considering the toxicity of solvent and limits approved? Would the dose of vancomycin being able to be delivered in a small solvent volume be adequate?
Author Response
Reply
Reply to Reviewer’s Comments
Manuscript ID: pharmaceutics-896123
Title: Saturated fatty acid-based in situ forming matrices for localised antimicrobial delivery via antisolvent process
Journal: Pharmaceutics (ISSN 1999-4923)
Firstly, we were grateful to receive all the valuable comments from the reviewers. These suggestions improve this manuscript. Responses to all the comments from the reviewers were explained as the following details point-by-point by as red and purple mark of the revised manuscript.
Reviewer 1 Comment:
Please see my comments below:
- Title can be clearer to read
:The title was changed to be concise and clearer as “Saturated Fatty Acid-based In Situ Forming Matrices for Localised Antimicrobial Delivery”
2. Abstract can be re-written to provide more quantitative information
:As your mentioned, the quantitative information was included in revised abstract as follows:
“In recent years, the world has faced the issue on antibiotic resistance. Methicillin-resistant Staphylococcus aureus (MRSA) is a significant problem in various treatments and control of infections. Biocompatible materials with saturated fatty acids of different chain lengths (C8–C18) were studied as matrix formers of localised injectable vancomycin HCl (VCM)-loaded antisolvent-induced in situ forming matrices. The series of fatty acid-based in situ forming matrices showed the low viscosity (5.47–13.97 cPs) and the pH value in the rank of 5.16-6.78 with high injectability through a 27-G needle (1.55–3.12 N). The preparations exhibited low tolerance to high concentrations of KH2PO4 solution (1.88-5.42 %v/v) and depicted an electrical potential change during phase transformation. Their phase transition and matrix formation at the microscopic and macroscopic levels with depended on the chain length of fatty acids and solvent characteristics. The VCM release pattern depended on the nucleation/crystallisation and solvent exchange behaviours of the delivery system. The 35% w/v of C12–C16 fatty acid-based in situ forming matrix prolonged the VCM release over seven days in which C12, C14, C16 –based formulation reached 56, 84 and 85 % cumulative drug release at 7th day. The release data fitted well with Higuchi's model. The developed formulations presented efficient antimicrobial activities against standard S. aureus, MRSA, Escherichia coli and Candida albicans. Hence, VCM-loaded antisolvent-induced fatty acid-based in situ forming matrix is a potential local delivery system for the treatment of local Gram-positive infection sites, such as joints, eyes, dermis of surgery sites, etc., in the future.
3. English needs to be carefully revised throughout the manuscript.
:The English of the content in manuscript carefully edited and checked as your concern and suggestion. The English in this manuscript was proven and language checked by native speaker and also Enago language check (1732, 1st Ave #22627, New York City New York, 10128 United States, T: +1-302-498-8358, E: orders@enago.com) and then revised.
4. Figure 1 is not providing really any benefit, please remove.
: As your mentioned, Figure 1 was deleted.
5. “Vancomycin HCl (VCM), a bulky hydrophilic drug (Fig. 1G), has been widely used to treat infections caused by Gram-positive bacteria, especially Staphylococcus aureus (including MRSA), S. epidermidis, Streptococcus spp., viridans streptococci and Enterococcus spp [28-31]”. Please state Ic50 against these species for vancomycin with appropriate references.
:The IC50 and MIC values of drug are informed in Introduction as follows:
“VCM has activity against Streptococcus spp., Enterococcus spp., Staphylococcus spp., and Clostidium spp. [28]. The half maximal inhibitory concentration (IC50)/minimum inhibition concentration(MIC) of vancomycin HCl against S. aureus, MRSA, and E. faecalis were 0.37/0.625, 0.67/0.125, 0.73/1.25 µg/mL, respectively [29]. While the MIC values of vancomycin HCl against beta-hemolytic streptococci, which include S. pyogenes, S. agalactiae, S. dysgalactiae, S. anginosus group, Group C streptococci and Group G streptococci were in the range of ≤0.015 to 1 µg/mL [30].” in page 2.
Ref:
Watanakunakorn, C. Mode of action and in-vitro activity of vancomycin. J. Antimicrob. Chemother. 1984, 14 Suppl D, 7-18.
Ahmed, M.H.; Ibrahim, M.A.; Zhang, J.; Melek, F.R.; El-Hawary, S.S.; Jacob, M.R.; Muhammad, I. Methicillin-resistant Staphylococcus aureus, vancomycin-resistant Enterococcus faecalis and Enterococcus faecium active dimeric isobutyrylphloroglucinol from Ivesia gordonii. Nat. Prod. Commun. 2014, 9, 221-224.
Dunne, M.W.; Sahm, D.; Puttagunta, S. Use of vancomycin as a surrogate for dalbavancin in vitro susceptibility testing: results from the DISCOVER studies. Ann. Clin. Microbiol. Antimicrob. 2015, 14, 19-19, doi:10.1186/s12941-015-0081-5.
6. VCM is the drug of choice for MRSA infection. – Please provide a reference to support this statement.
:The following cited references was added for this statement.
31. Vazquez-Guillamet, C.; Kollef, M.H. Treatment of gram - positive infections in critically ill patients. BMC Infect. Dis. 2014, 14, 92, doi:10.1186/1471-2334-14-92.
- Micromedex. Vancomycin. Thomson Micromedex.: Greenwood Village, CO, 2017.
- Roy, M.E.; Peppers, M.P.; Whiteside, L.A.; LaZear, R.M. Vancomycin Concentration in Synovial Fluid: Direct Injection into the Knee vs. Intravenous Infusion. J. Arthroplasty 2014, 29, 564-568, doi:10.1016/j.arth.2013.08.017.
7. Please state purity of all lipids, solvents and materials.
:The purity of all materials was informed as follows:
“CPL (≥99%, Batch No. 289D1190515A), CPR (≥99%, Batch No. 219D161222A), LAU (≥99%, Batch No. 229F170S08A), MYR (≥99%, Batch No. FPLK437X4S) and PAL (≥99%, Batch No. 668F180902D), which were procured from Pacific Oleochemicals, Pasir Gudang Johor Darul Takzim, Malaysia and STR (≥98, Batch No. 3452018), which was purchased from CT Chemical, Bangkok, Thailand, were used as matrix formers. VCM (≥80%, Lot No. WXBB5169V, Sigma-Aldrich, Co., USA) was used as a model drug. NMP (≥99.5%, Lot No. A0251390, Fluka, New Jersey, USA) and DMSO (≥99.9%, Lot No. 453035, Fluka, Switzerland) were used as solvents. Muller Hinton agar (Lot No. 1005843, Oxoid, UK) was used as medium for the antimicrobial tests. S. aureus ATCC 43300, S. aureus DMST 6532, S. aureus ATCC 25923, E. coli ATCC 8739 and C. albicans were used as test microbials. Potassium dihydrogen orthophosphate (≥99%, Lot No. E23W60, Ajax Finechem, Australia) and sodium hydroxide (≥97%, Lot No. AF 310204, Ajax Finechem, Australia) were used as components of phosphate buffer (PB) with pH 7.4 to simulate the physiological fluid. Agarose (Lot No. H7014714, Vivantis, Malaysia) and sodium–fluorescein (Lot No. MXCG7851, Sigma-Aldrich, USA) were used for the analysis of matrix formation behaviour and solvent diffusion.” in page 3.
- How was the pH of the formulation evaluated in a non-aqueous solvent?
:The pH meter (Seven Compact, METTLER TOLEDO, USA) was used to measure the pH value. Typically, the pH measurement in the non-aqueous condition is not preferred; nevertheless, the comparison of the pH value is possible for non-aqueous mixture in the formulation series. Many research and IUPAC are also interested in this issue (Rondinini 2002). The authors understand with your concern; nevertheless, the pH value of formulation (apparent pH) was often asked and the result of each sample had low standard deviation. Thus, the pH value of our formulation is informed as shown in Table 2 which the obtained data was applicable for discussion.
Ref: Rondinini, S. (2002). "pH measurements in non-aqueous and aqueous–organic solvents – definition of standard procedures." Analytical and Bioanalytical Chemistry 374(5): 813-816.
9. For conductomer experiments, please state values obtained for water as a standard.
The value obtained for water was state as follows:
:“The voltage of deionized water was 371.2±1.3. mV with conductivity of 0.1µs/cm.” This information is added in topic 2.3.5 Electrical potential difference during phase transformation as your suggestion.
10. Please state retention time obtain using HPLC method and lower limit of detection.
:The retention time obtain using HPLC method and lower limit of detection were included and stated in topic: 2.3.8 In vitro drug release as follows:
“The retention time of VCM was 7 min and the limit of detection was 1.1524 μg/mL (from Equation 2).”
….Eq. 2
“Where σ is the standard deviation of the response and S is the slope of the calibration curve.”
11. The 35% w/v fatty 218 acid solutions showed an increased pH when a high-MW fatty acid was used due to the decrease in fatty acid molecules with a carboxyl group. Shouldn’t the opposite be true? The higher the mw of fatty acid, the lower the moleculers, thus lower content of carboxylic acids?
:For example, the 35%w/v of Lauric acid has 1.75*6.02*1023 lauric acid molecules per 1 liter and 35% w/v of Palmitic acid has 1.36*6.02*1023 palmitic acid molecules per 1 liter. Nevertheless, the authors revised this sentence to following sentence:
:“The 35% w/v fatty acid solutions showed an increased pH when a higher-MW fatty acid was used due to the less fatty acid molecules.” in topic 3.1 pH, density and viscosity
12. I am not really certain about validity of pH measurements within pure solvent mixtures? What does that really show?
:Since the pH value of formulation (Apparent pH) was often asked from the reader; thus, we had determined the pH value of our formulation and the result of each sample exhibited low standard deviation. The pH meter (Seven Compact, METTLER TOLEDO, USA) was used to measure the pH value. Typically, the pH measurement in the non-aqueous condition is not preferred; nevertheless, the comparison of the pH value is possible for non-aqueous mixture in the formulation series. Many research and IUPAC are also interested in this issue (Rondinini 2002). The authors understand with your concern; however, usually the pH value of formulation (apparent pH) was needed and the result of each sample had low standard deviation. Thus, the pH value of our formulation is also informed as shown in Table 2 which the obtained data was applicable for discussion and showed the informative relationship with fatty acid character as described in page 8-9 in topic 3.1 and Table 2.
Ref: Rondinini, S. (2002). "pH measurements in non-aqueous and aqueous–organic solvents – definition of standard procedures." Analytical and Bioanalytical Chemistry 374(5): 813-816.
13. Lines 236-253 are unclear. Cannot follow clearly the arguments made and are confused. Please re-write.
:To solve the unclear point of view and more understanding, the re-write is conducted as following paragraphs:
“Moreover, according to the viscosity results, the trend of viscosity change was similar to that of density. It was reported that these two values are related [54].
“All the solutions had low viscosity given that a simple and small structure of fatty acid can lessen the inter/intra-molecular interaction, different from bulky molecules such as polymer. Eudragit® RS, a gel former of recently developed in situ forming gel, can readily interact with vehicles, resulting in a steep viscous preparation when the concentration was increased [55]. Comparison between the formulations, a higher viscosity was associated with the use of a higher MW fatty acid. In the study of binary liquid mixtures of fatty acids, the solutions exhibited an increased viscosity when the ratio of long-chain fatty acid was increase [56]. The following reason might explain for the increase of the viscosity. Firstly, the interaction from carboxyl group via hydrogen bonding. The functional group as carboxyl was perceived for behaviour of fatty acids in solutions. It was reported that the carboxyl group of fatty acid protruding from the cavity of amylose–lipid complexes can interact with other molecules, resulting in increased viscosity [57,58]. Although the hydrogen bonding between carboxylic groups can typically promote viscosity [59], the hydrophobic interaction should be decisively considered. The hydrophobic interaction can enhance viscosity [60]; the long chain length of alkanes induces the increase in hydrophobic interaction, resulting in the high viscosity of silicone oil [61]. However, the results from the 35% w/v fatty acid series might not be distinct since the less number of fatty acid molecules were evident for high MW fatty acid-based solutions.”
References:
Rodenbush, C.M.; Hsieh, F.H.; Viswanath, D.S. Density and viscosity of vegetable oils. J. Am. Oil Chem. Soc. 1999, 76, 1415-1419.
Phaechamud, T.; Thurein, S.M.; Chantadee, T. Role of clove oil in solvent exchange-induced doxycycline hyclate-loaded Eudragit RS in situ forming gel. Asian J. Pharm. 2018, 13, 131-142, doi:10.1016/j.ajps.2017.09.004.
Iwahashi, M.; Takebayashi, S.; Taguchi, M.; Kasahara, Y.; Minami, H.; Matsuzawa, H. Dynamical dimer structure and liquid structure of fatty acids in their binary liquid mixture: decanoic/octadecanoic acid and decanoic/dodecanoic acid systems. Chem. Phys. Lipids 2005, 133, 113-124, doi:10.1016/j.chemphyslip.2004.09.022.
Raphaelides, S.N.; Georgiadis, N. Effect of fatty acids on the rheological behaviour of amylomaize starch dispersions during heating. Food Res. Int. 2008, 41, 75-88, doi:10.1016/j.foodres.2007.10.004.
Meng, S.; Ma, Y.; Sun, D.-W.; Wang, L.; Liu, T. Properties of starch-palmitic acid complexes prepared by high pressure homogenization. J. Cereal. Sci. 2014, 59, 25-32, doi:10.1016/j.jcs.2013.10.012.
Small, D.M. The physical chemistry of lipids : from Alkanes to phospholipids. In Handbook of lipid research, Plenum Press: New York, 1986; Vol. 4, p. 264.
Viken, A.L.; Spildo, K.; Reichenbach-Klinke, R.; Djurhuus, K.; Skauge, T. Influence of Weak Hydrophobic Interactions on in Situ Viscosity of a Hydrophobically Modified Water-Soluble Polymer. Energy Fuels 2018, 32, 89-98, doi:10.1021/acs.energyfuels.7b02573.
Liang, Y.; Yuan, X.; Wang, L.; Zhou, X.; Ren, X.; Huang, Y.; Zhang, M.; Wu, J.; Wen, W. Highly stable and efficient electrorheological suspensions with hydrophobic interaction. J. Colloid Interface Sci. 2020, 564, 381-391, doi:https://doi.org/10.1016/j.jcis.2019.12.129.
14. Table 2 correct Dengityg
:The “Densityg” was changed into “Density”
15. Statistics for Figure 3 are missing
:The statistical analysis with statistic marks was included in Figure 3 (Recounted to Fig.2 since the Fig.1 was deleted as your mentioned)
16. Little evidence of crystallinity is provided although discussed. Not sure how provided data support this discussion.
: Practically, the matrix formation of fatty acid was owing to phase separation with its crystallization. These fatty acid crystals were obtained from the in situ forming process, their crystallinity manner might be different from those pure compounds or crystals obtained from another precipitation method. Nevertheless, the matrix formation with fatty acid crystal growth from Aqueous phase influx tracking is described in topic 3.9, page 18 and photomicrograph SEM from topography study indicated the crystal formation from tested fatty acids is also described in topic 3.9, page 20. The profound further investigation on crystallinity will be also further reported with dynamic computerized modelling of these fatty acid in situ forming matrices.
17. Is unionized water – deionized? Figure 5
:Yes, as your mentioned; thus, the “unionized water” is changed to “deionized water” in these figure and Method.
18. Figure 6 a and b – statistics are missing
:The statistical analysis with statistic marks are included in Figure 6 a, b (Recounted to Fig.5 a,b since the Fig.1 was deleted)
19. Figure 8 – Is quantification possible using Image J?
: The indication for boundary and some informative points of crystal growth in this Figure were included as your suggestion.
20. Figure 9- Calculate porosity using Image J
: Porosity of VPD and VLD matrices were measured using JMicroVision. The obtained data was included and discussed:
“The VLD dried matrix presented a needle-shaped crystal agglomerate with a continuously intricated matrix (average porous size = 4.205 μm). However, VPD showed an enclosed sheath crystal with a large and simple aperture (average porous size = 6.376 μm).”
21. Would this in situ gel be able to be clinically translated and if so which application is envisaged considering the toxicity of solvent and limits approved? Would the dose of vancomycin being able to be delivered in a small solvent volume be adequate?
:The clinical translation could be engaged. This in situ forming system could be used at various local site where the application of formulation was not exceeded 15 mL (regarding to the US-FDA limit of DMSO is 104 mg/dose via subcutaneous). The developed system was loaded quite high amount of vancomycin HCl where the concentration was high enough for inhibition against the MRSA. For example, knee joint (in the case of post-operative knee infection), only 0.5-3.0 mL injection volume of in situ forming system is filled into a 5-30 mL synovial fluid. Regarding to pharmacokinetic, vancomycin concentration in knee joint is estimated the as follows:
Since the minimal inhibitory concentration of vancomycin against MRSA is 2-4 μg/mL and the elimination half-life of Intra-articular vancomycin is 3.22 hours (Roy, Peppers et al. 2014, Whiteside, Roy et al. 2016). The estimated concentration in this table were calculated from the equation given below based on following conditions: the synovial fluid volume of 30 mL, elimination half-life 3.22 hr-1, 3 mL of VMD injection (30,000 µg of Vancomycin HCl).
, Ct = Concentration at given time (μg/ml)
ke = elimination rate constant (hr-1), 0.693/t1/2: 0.693/3.22
t = Given time
t-1=Previous time point
Table. 1 Estimate concentration of Vancomycin HCl from VMD in knee joint with 30 mL synovial fluid
|
|
A |
B |
C |
D |
E |
|
Time (hr) |
Cumulative release of VCM (ug/ml) |
ΔRelease |
Ct (ug/ml) |
Estimate conc. of VCM in knee joint (ug/ml), A+C |
Area under curve, 0.5x(Dt+Dt-1)x(Δt) |
|
0 |
0 |
0 |
0 |
0 |
0 |
|
0.5 |
50.88 |
50.89 |
0 |
50.88759242 |
12.72 |
|
1 |
96.24 |
45.36 |
45.70 |
91.0582978 |
35.49 |
|
2 |
150.96 |
54.72 |
73.43 |
128.1446686 |
109.60 |
|
4 |
308.63 |
157.66 |
83.32 |
240.9857427 |
369.13 |
|
6 |
316.01 |
7.377 |
156.70 |
164.0725585 |
405.06 |
|
12 |
476.35 |
160.34 |
45.11 |
205.4493798 |
1108.57 |
|
24 |
484.44 |
8.08 |
15.53 |
23.6111983 |
1374.36 |
|
48 |
600.61 |
116.18 |
0.13 |
116.3124428 |
1679.08 |
|
72 |
808.47 |
207.85 |
0.66 |
208.5175572 |
3897.96 |
|
120 |
825.98 |
17.51 |
0.01 |
17.52106345 |
5424.93 |
|
144 |
847.56 |
21.58 |
0.10 |
21.67967876 |
470.41 |
The results showed the high concentration vancomycin HCl over MIC level at every time point. Moreover, based on the Pharmacokinetic-Pharmacodynamics criteria, AUC/MIC ratio of ≥400 has been predicted value as a target to achieve clinical effectiveness with vancomycin for bacteremia (Prybylski 2015, Men, Li et al. 2016), this estimate calculation indicate AUC/MIC of 3721.8 μg/ml.
Ref:
Roy, M. E., M. P. Peppers, L. A. Whiteside and R. M. LaZear (2014). "Vancomycin Concentration in Synovial Fluid: Direct Injection into the Knee vs. Intravenous Infusion." The Journal of Arthroplasty 29(3): 564-568.
Whiteside, L. A., M. E. Roy and T. A. Nayfeh (2016). "Intra-articular infusion." The Bone & Joint Journal 98-B(1_Supple_A): 31-36.
The authors add the more information about the point of view as following:
“In order to clinically translate, the limit of DMSO and pharmacokinetic of VCM depending on an application site should be considered. DMSO has been used as a solvent of various dosage forms such as risperidone and paliperidone-loaded in situ forming implants [11], osthole-loaded in situ forming implants [91], in situ forming microparticle loading a montelukast sodium [92], and rapamycin-loaded intra-arthricular injection [93]. US FDA has approved to use 50%w/w DMSO (RIMSO-50®) for human interstitial/chronic cystitis treatment as well as various topical dosage form containing DMSO for flap ischemia (60% DMSO), herpes zoster (5% idoxuridine in DMSO), and injection site extravasation (99%DMSO) treatment [94]. Although the safety data and applications of DMSO indicate the probability of using DMSO as the solvent for developed dosage form, the effect of DMSO from the formulation such as the developed VLD, VMD and VPD need to be determined more in a clinical experiment. For example, in the case of post-operative knee infection, cell recovery with toxicity kinetic study of DMSO in an intact articular cartilage, the 1 M DMSO did not cause a toxic to the chondrocytes [95]. The estimate DMSO concentration after injection of 0.5 - 3.0 mL of developed formulation into 5-30 mL synovial fluid was in the rank of 0.004-0.025 M and the estimated concentration of VCM were above MRSA MIC level over 6 day, based on 3.22 hr-1 VCM half-life in synovial fluid and 2-4 μg/mL MRSA MIC level [33,96].” In second paragraph of page 22.
However, this study aims to provide the fundamental data to design a proper fatty acid-based in situ forming matrix for any aspect of drug delivery applications. The further investigation which suit to any clinical application site should be conducted efficiently.
References:
Wang, L.; Wang, A.; Zhao, X.; Liu, X.; Wang, D.; Sun, F.; Li, Y. Design of a long-term antipsychotic in situ forming implant and its release control method and mechanism. Int. J. Pharm. 2012, 427, 284-292, doi:10.1016/j.ijpharm.2012.02.015.
Roy, M.E.; Peppers, M.P.; Whiteside, L.A.; LaZear, R.M. Vancomycin Concentration in Synovial Fluid: Direct Injection into the Knee vs. Intravenous Infusion. J. Arthroplasty 2014, 29, 564-568, doi:10.1016/j.arth.2013.08.017.
Zhang, X.; Yang, L.; Zhang, C.; Liu, D.; Meng, S.; Zhang, W.; Meng, S. Effect of Polymer Permeability and Solvent Removal Rate on In Situ Forming Implants: Drug Burst Release and Microstructure. Pharmaceutics 2019, 11, 520, doi:10.3390/pharmaceutics11100520.
Ahmed, T.A.; Ibrahim, H.M.; Samy, A.M.; Kaseem, A.; Nutan, M.T.H.; Hussain, M.D. Biodegradable Injectable In Situ Implants and Microparticles for Sustained Release of Montelukast: In Vitro Release, Pharmacokinetics, and Stability. AAPS. J. 2014, 15, 772-780, doi:10.1208/s12249-014-0101-3.
Takayama, K.; Kawakami, Y.; Kobayashi, M.; Greco, N.; Cummins, J.H.; Matsushita, T.; Kuroda, R.; Kurosaka, M.; Fu, F.H.; Huard, J. Local intra-articular injection of rapamycin delays articular cartilage degeneration in a murine model of osteoarthritis. Arthrit. Res. Ther. 2014, 16, 482-482, doi:10.1186/s13075-014-0482-4.
Micromedex. Dimethyl Sulfoxide. Thomson Micromedex.: Greenwood Village, CO, 2020.
Elmoazzen, H.Y.; Poovadan, A.; Law, G.K.; Elliott, J.A.W.; McGann, L.E.; Jomha, N.M. Dimethyl sulfoxide toxicity kinetics in intact articular cartilage. Cell Tissue Banking 2006, 8, 125, doi:10.1007/s10561-006-9023-y.
Whiteside, L.A.; Roy, M.E.; Nayfeh, T.A. Intra-articular infusion. The Bone & Joint Journal 2016, 98-B, 31-36, doi:10.1302/0301-620X.98B.36276.
Reviewer 2 Report
The article reports the influence of C-chains lenght on matrices formation for applications as drug delivery systems with antimicrobial activity.
In general, the article is well-designed and results are significant, considering the different characterization techniques in addition to in vitro biological activities. However, a few points should be considered by authors:
In vitro release studies: please provide data analysis according to mathematic models to investigate if there are differences on release mechanisms according to fatty acid structure.
Is it possible to stablish a structure-biological activity relationship for the different matrices ? Please discuss.
Rheological studies are suggested, since a more detailed discussion about matrices mechanical properties are necessary.
Author Response
Reply to Reviewer’s Comments
Manuscript ID: pharmaceutics-896123
Title: Saturated fatty acid-based in situ forming matrices for localised antimicrobial delivery via antisolvent process
Journal: Pharmaceutics (ISSN 1999-4923)
Firstly, we were grateful to receive all the valuable comments from the reviewers. These suggestions improve this manuscript. Responses to all the comments from the reviewers were explained as the following details point-by-point by as red or purple mark of the revised manuscript.
Reviewer 2 Comment:
In general, the article is well-designed and results are significant, considering the different characterization techniques in addition to in vitro biological activities. However, a few points should be considered by authors:
- In vitro release studies: please provide data analysis according to mathematic models to investigate if there are differences on release mechanisms according to fatty acid structure.
:The data analysis according to mathematic models was provided as your suggestion. The following paragraphs were added into the manuscript:
“Considerable release models with different mathematical equations for this experimental data, including power-law, zero-order, first-order, and Higuchi’s were used for drug release profile fitting. The coefficient of determination (r2) was the parameter for indicating the degree of curve fitting. According to the Korsmeyer-Peppas model, the diffusion exponent (n) from the power-law model indicates the drug transport mechanism”
“The estimated r2 from the VCM cumulative release data fitted to different mathematical release models is shown in Table 3. All release profiles fitted well with Higuchi’s equation which depicted the migration of VCM through the tortuosity matrix in which the distance from the matrix surface to the drug dissolution region was accounted [81,82]. For the non-swelling system with a cylindrical shape, the diffusion exponent (n) implies the transport of drug, where n=0.89 indicates the case II transport mechanism, n=0.45 indicates Fickian diffusion, and 0.45<n<0.89 indicates anomalous diffusion. The Fickian diffusion refers to the drug transport mechanism in which the diffusion rate is faster than that of structure relaxation. Whereas, when the relaxation process is higher than diffusion, it is Case II transport. Otherwise, when these rates are close to each other, anomalous diffusion is acquired [44,45,83,84]. According to those cited, the VCM was released from VLAUD and VMYRD via the mechanism of anomalous diffusion while VPALD presented a pure Fickian diffusion.” in page 17.
Table 4 Degrees of goodness-of-fit and estimated parameters from curve fittings of the release profiles of VCM-loaded in situ forming systems in phosphate buffer at pH 7.4 using the membrane-less method
|
|
Zero-order |
First-order |
Higuchi’s |
Korsmeyer-peppas |
|||||
|
|
r2 |
k |
r2 |
k |
r2 |
k |
r2 |
k |
n |
|
VLAUD |
0.8374 |
0.3113 |
0.5863 |
0.0056 |
0.9584 |
4.4057 |
0.9593 |
0.0498 |
0.5062 |
|
VMYRD |
0.8130 |
0.7792 |
0.5470 |
0.0088 |
0.9451 |
8.3685 |
0.9334 |
0.1048 |
0.4923 |
|
VPYRD |
0.7537 |
0.7772 |
0.5576 |
0.0082 |
0.9169 |
8.5386 |
0.9471 |
0.1239 |
0.4589 |
r2=coefficient of determination, k=constant, n= diffusion exponent.
Ref:
Ritger, P.L.; Peppas, N.A. A simple equation for description of solute release I. Fickian and non-fickian release from non-swellable devices in the form of slabs, spheres, cylinders or discs. J. Control. Release 1987, 5, 23-36, doi:10.1016/0168-3659(87)90034-4.
Costa, P.; Sousa Lobo, J.M. Modeling and comparison of dissolution profiles. Eur. J. Pharm. Sci. 2001, 13, 123-133, doi:https://doi.org/10.1016/S0928-0987(01)00095-1.
Higuchi, T. Rate of Release of Medicaments from Ointment Bases Containing Drugs in Suspension. J. Pharm. Sci. 1961, 50, 874-875, doi:https://doi.org/10.1002/jps.2600501018.
Higuchi, T. Mechanism of sustained‐action medication. Theoretical analysis of rate of release of solid drugs dispersed in solid matrices. J. Pharm. Sci. 1963, 52, 1145-1149, doi:10.1002/jps.2600521210.
Thomas, N.L.; Windle, A.H. A theory of case II diffusion. Polymer 1982, 23, 529-542, doi:https://doi.org/10.1016/0032-3861(82)90093-3.
M. Padmaa, P.; Preethy Ani, J.; Cm, S.; G.V. Peter, C. Release kinetic n concept and applications. IJPRT 2018, 8, 12.
- Is it possible to stablish a structure-biological activity relationship for the different matrices? Please discuss.
: Your suggested possibility is described from the authors by inserting following content to point the this relationship.
“This section results depicted not only the relation between antimicrobial activity and fatty acid type as mentioned details but also the formed matrix structure. As seen in the previous section, the structure of VLD and VPD matrices are distinct. The higher porosity, smaller pieces, and needle shape crystals of VLD formulation might relate to the higher antimicrobial activity. The relation between these characters and antimicrobial activity was also mentioned in other studies where the high porous and small particle size could enhance antimicrobial activity [89,90].”
References:
Talebian, N.; Amininezhad, S.M.; Doudi, M. Controllable synthesis of ZnO nanoparticles and their morphology-dependent antibacterial and optical properties. J. Photochem. Photobiol. B: Biol. 2013, 120, 66-73, doi:https://doi.org/10.1016/j.jphotobiol.2013.01.004.
Pasquet, J.; Chevalier, Y.; Couval, E.; Bouvier, D.; Noizet, G.; Morlière, C.; Bolzinger, M.-A. Antimicrobial activity of zinc oxide particles on five micro-organisms of the Challenge Tests related to their physicochemical properties. Int. J. Pharm. 2014, 460, 92-100, doi:https://doi.org/10.1016/j.ijpharm.2013.10.031.
Rheological studies are suggested, since a more detailed discussion about matrices mechanical properties are necessary.
:From rheological determination, all these unique low viscous formula exhibited Newtonian flow behavior with low viscosity value as presented in Table 2 and high injectability character. The type of rheological behavior of the formulations was added in the manuscript as follows:
“The rheological behaviour was determined using a Brookfield DV-III Ultra programmable rheometer (Brookfield Engineering Laboratories Inc, Middleboro, MA, USA) (n=3). The shear stress of samples was measured at various shear rates at 25°C. The flow parameters were characterized using an exponential formula with N as the exponential constant (Farrow's constant) and η´ as the viscosity coefficient.” in page 6 of topic 2.3.7
“Moreover, all of the formulations in fatty acid-based series exhibited Newtonian flow behavior in which the flow index (N) was not statistic different. Even the lowest injectability as 1.75SN, a Newtonian behavior where obtained (N=1.000±0.006). This flow behaviour was an expected pattern in an injection dosage form [78].” in page 16-17.
Reference:
Elnaggar, Y.S.R.; El-Refaie, W.M.; El-Massik, M.A.; Abdallah, O.Y. Lecithin-based nanostructured gels for skin delivery: An update on state of art and recent applications. J. Control. Release 2014, 180, 10-24, doi:10.1016/j.jconrel.2014.02.004.
Round 2
Reviewer 1 Report
Accept as is
Reviewer 2 Report
The authors revised the manuscript and considered all suggestions.